# Design and Application of Near-Infrared Nanomaterial-Liposome Hybrid Nanocarriers for Cancer Photothermal Therapy

**DOI:** 10.3390/pharmaceutics13122070

**Published:** 2021-12-03

**Authors:** Pan Liang, Linshen Mao, Yanli Dong, Zhenwen Zhao, Qin Sun, Maryam Mazhar, Yining Ma, Sijin Yang, Wei Ren

**Affiliations:** 1National Traditional Chinese Medicine Clinical Research Base and Drug Research Center of the Affiliated Traditional Chinese Medicine Hospital of Southwest Medical University, Luzhou 646000, China; xnydzyylp@swmu.edu.cn (P.L.); maolinshen@swmu.edu.cn (L.M.); 20190399120004@stu.swmu.edu.cn (Y.D.); zxyjhsq@swmu.edu.cn (Q.S.); maryam@swmu.edu.cn (M.M.); 20210399120027@stu.swmu.edu.cn (Y.M.); 2College of Integrated Traditional Chinese and Western Medicine, Southwest Medical University, Luzhou 646000, China; 3Beijing National Laboratory for Molecular Sciences, CAS Research/Education Center for Excellence in Molecular Sciences, Key Laboratory of Analytical Chemistry for Living Biosystems, Institute of Chemistry Chinese Academy of Sciences, Beijing Mass Spectrum Center, Beijing 100190, China; zhenwenzhao@iccas.ac.cn

**Keywords:** near-infrared nanomaterials, liposomes, hybrid nanocarriers, targeted drug delivery, photothermal therapy

## Abstract

Liposomes are attractive carriers for targeted and controlled drug delivery receiving increasing attention in cancer photothermal therapy. However, the field of creating near-infrared nanomaterial-liposome hybrid nanocarriers (NIRN-Lips) is relatively little understood. The hybrid nanocarriers combine the dual superiority of nanomaterials and liposomes, with more stable particles, enhanced photoluminescence, higher tumor permeability, better tumor-targeted drug delivery, stimulus-responsive drug release, and thus exhibiting better anti-tumor efficacy. Herein, this review covers the liposomes supported various types of near-infrared nanomaterials, including gold-based nanomaterials, carbon-based nanomaterials, and semiconductor quantum dots. Specifically, the NIRN-Lips are described in terms of their feature, synthesis, and drug-release mechanism. The design considerations of NIRN-Lips are highlighted. Further, we briefly introduced the photothermal conversion mechanism of NIRNs and the cell death mechanism induced by photothermal therapy. Subsequently, we provided a brief conclusion of NIRNs-Lips applied in cancer photothermal therapy. Finally, we discussed a synopsis of associated challenges and future perspectives for the applications of NIRN-Lips in cancer photothermal therapy.

## 1. Introduction

Currently, cancer treatment mainly relies on chemotherapeutic drugs [1]. However, after administration, free drugs are also trapped in the normal tissue because of the low selectivity for tumor tissue [2]. Nanomedicine is a rising field that possesses the main advantages of effectively targeted drug delivery to the tumor cells through enhanced permeation and retention (EPR) effects of nanocarriers [3]. In fact, it is a hard process for conventional nanomedicine to deliver drugs to tumor tissue, which must undergo five steps such as circulation, accumulation, penetration, internalization, and release [4]. Liposomes were widely studied as the potential drug delivery system in the early 1970s [5,6]. This nanocarrier possesses man-made spherical nanostructures with an aqueous inner core surrounded by phospholipid bilayers, similar to cell membranes [7]. From theoretical concept to clinical practice, liposomes were the first drug delivery system, which were designed to improve therapeutic effects of various water-soluble and insoluble drugs through enhancing bioavailability, solubility, retention time and reducing systemic toxicity [8,9]. Different drug molecules can be loaded into liposomes by hydrophilic–hydrophilic and hydrophobic–hydrophobic interactions [10]. Nevertheless, it is far from satisfactory in terms of drug concentration and delivery efficiency at the target site at the right time [11]. Moreover, conventional liposomes have limited clinical applications because of their inherent physicochemical instability, uncontrollable release of encapsulated drugs, and short circulation lifetime in vivo [12]. Therefore, there is an urgent need to develop smart liposome-based delivery systems with the ability to control the release of therapeutic payloads in response to a given stimulus [13]. 

Specifically, external stimuli (such as light, temperature, and ultrasound) are perceived by sensing elements in the liposomes, thus destabilizing the liposomes and leading to the controlled release of liposomal encapsulated drugs [11]. Among them, owing to its non-invasiveness, ease of application, and fine temporal and spatial control, light is a particularly attractive tool for on-demand drug delivery [14]. In comparison to other stimuli such as temperature and ultrasound, light has superiorities of exhibiting no side effects to the biological environment and allowing precisely targeted drug delivery with high spatio-temporal resolution in non-contact mode, thereby improving treatment effect and patient comfort [14,15,16]. Light with various wavelengths can be used to activate the light response, ranging from ultraviolet (UV, 200–400 nm) to visible light (400–750 nm) or near-infrared (NIR, 750–2000 nm) [14]. UV light is a relatively poor candidate because of its limited tissue penetration and potential carcinogenic effects under long-term exposure [16,17]. In contrast, NIR light with characteristics of lower phototoxicity, increased penetration depth of biological tissues, and weakened background signals, is more suitable for biological applications [18]. Over the past decade, the development of photothermal therapy (PTT) has attracted significant research interest in combating different types of cancers and malignant tumors [19]. PTT applied in tumor ablation possesses several advantages, including minimal invasiveness, deeper tissue penetration by NIR light, better tumor specificity, less systemic toxicity, and spatiotemporally controlled drug release [20,21]. This therapeutic strategy mainly relies on the absorption of light by photothermal agents (PTAs) to convert the absorbed photon energy into heat, resulting in rapid enhancement of the local cell temperature to 42–45 °C in 15–60 min to ablate cancer cells [21,22]. PTAs should have the characteristics of low or non-toxicity, strong NIR absorption and high photothermal conversion efficiency [23]. In fact, most PTAs can absorb light in the visible range (wavelength of 400–750 nm) or NIR range (wavelength of 750–1350 nm) [24]. In general, the wavelength of 650–900 nm NIR light is more suitable for cancer PTT due to the weaker tissue absorption and deeper tissue penetration [25].

A greater variety of PTAs have been extensively developed based on small organic molecules (indocyanine green, IR780, and IR825) and inorganic nanomaterials, which are available in several excellent reviews [26,27,28]. Indocyanine green (ICG) is an NIR fluorescent dye that was approved by the FDA in 1959, which is also a promising PTA under in-depth preclinical and clinical investigation [29]. Despite its excellent optical properties, the application of ICG in PTT is limited by its short plasma residence time (about 2–4 min), rapid photo-degradation in aqueous solution, and concentration-dependent peak emission location [30,31]. It has previously been shown that the phototoxicity of ICG under light exposure is mainly due to the presence of dissolved photodegradation products in the cytoplasm rather than the generation of singlet oxygen [32]. Owing to their excellent imaging capacity and photothermal conversion efficiency, inorganic PTAs are currently prioritized by researchers to apply in cancer diagnosis and treatment [33]. In particular, NIR light-activated inorganic nanomaterials (NIRNs) possess unique shape and size-dependent optical and electrical properties, and large surface areas, which create a great promise for their cancer PTT [14]. Furthermore, the spectral absorption peak in the NIR band greatly increases the transmittance depth and reduces light absorption by tissues [21]. Nevertheless, the potential toxicity and low biocompatibility characteristics of NIRNs may pose risks to the body [14,28]. In comparison, liposomes have good biocompatibility and easy biodegradability. Therefore, many researchers have focused on the utilization of NIRNs-liposome hybrid nanocarriers (NIRN-Lips) as they have been observed to overcome drawbacks of NIRNs [34,35,36,37].

This review mainly focuses on the recent advances in liposomes loading various NIRNs for cancer PTT, including carbon-based nanomaterials, gold-based nanomaterials, and semiconductor quantum dots (Figure 1). Specifically, the rational design and surface modification for the fabrication of NIRN-Lips are discussed. The effects of size, shape, and surface functionalities of NIRN-Lips on the antitumor activities in vitro and in vivo are also reviewed. Then, we briefly discuss the photothermal conversion mechanism of NIRNs and NIR light triggered drug release from NIRN-Lips. An overview of the recently developed NIRN-Lips for cancer PTT is presented, especially for the combination of NIRN-Lips with chemotherapy. Further, associated challenges and future perspectives in this rapidly growing field will be discussed.

## 2. Types of NIRN-Lips

### 2.1. Liposomes Loading Carbon-Based Nanomaterials

Carbon-based nanomaterials such as graphene oxide (GO), carbon nanotubes (CNTs), carbon nanohorns (CNHs), and carbon dots (CDs), have been extensively studied as photothermal transduction agents due to their elongated conjugate bands with strong NIR light absorption [19]. GO is a single-layer graphite with unique thermal, electrical, and optical properties, which exhibits remarkably photothermal conversion through the plasma photothermal effect [38]. In comparison with CNTs, GO has a better photothermal conversion effect, which is closely linked to its two-dimensional structural property [39]. The ability of GO to cause local heating makes it a promising candidate for cancer PTT [40]. CNTs, the rolled graphene sheets (diameter of 1–100 nm) with remarkable physical and mechanical properties, have received great interest in antitumor drug delivery [41]. In particular, multi-walled carbon nanotubes (MWNTs) have become the most promising carbon-based nanomaterials in cancer therapy due to their exceptionally large surface areas [42]. It allows multiple molecules to be efficiently loaded along the wall of the nanotube through the supramolecular stacking π–π interaction [43]. In the past decade, due to the strong optical absorption of CNTs under NIR light, studies on different CNTs-based PTT of cancer in animal models have been reported [44,45]. Generally, anticancer drugs can be attached to CNTs through covalent or non-covalent interactions between drugs and functionalized CNTs [46]. The structure of CNHs is similar to CNTs, which is a horn-shaped nanomaterial with 2–5 nm in width and 40–50 nm in length [47]. Notably, single-walled carbon nanohorns (SWNHs) usually aggregate into flower-like spherical clusters with a diameter of 80–120 nm and provide a large hydrophobic surface for drug loading [48]. Moreover, SWNHs are optimal for PTT of deeper tissues without the interference of photon scattering due to their broad absorbance in both NIR I and II areas [49,50]. In comparison to SWNTs and GO, the non-toxic production process and unique morphology enable SWNHs to have different nano-protein and nano-membrane interaction patterns [51]. Like CNTs and GO, CNHs also lack solubility in aqueous media, which seriously hampered their practical application [52,53]. Unlike the previously mentioned carbon-based nanomaterials, CDs with sizes less than 10 nm and ideal water solubility, as the fourth type of carbon-based nanomaterials, have received considerable attention since they were firstly reported in 2004 [54]. The optical stability, quantum yield, and water solubility of CDs are better than those of quantum dots [54]. The fluorescence of CDs was derived from the nuclear and surface states of carbon sources, the properties of which could be adjusted by changing the surface functional groups of CDs [55]. Targeting molecules and antitumor drugs can be jointly modified to the large surface of CDs, helping to identify tumor sites and release drugs [56].

Encapsulating carbon-based nanomaterials into liposomes have beneficial effects on PTT. For example, Mohadeseh Hashemi et. al. reported a dual chemophotothermal therapeutic platform prepared by the deposition of GO on the surface of cationic liposomes [57]. The GO on the liposome acts as a photothermal sensor (NIR absorber and heating agent). The infrared reaction shows that the four-layer GO is sufficient to induce the gel-liquid phase transition of liposomes under NIR laser irradiation and trigger the release of the encapsulated toxic cargos. Moreover, in our previous work, we investigated the feasibility of NIR fluorescent CDs-encapsulated liposomes as multifunctional nanocarriers, in vivo and in vitro tracers, and efficacy enhancers for the anticancer agent cinobufagin (CB). As shown in Figure 2A, by embedding CDs into liposomes, the photoluminescence intensity of CDs was increased by 5-fold, which is helpful for tumor NIR imaging. This hybrid liposome system has multiple functions, including tumor-targeted drug delivery and imaging, sustained drug release, enhanced photoluminescence emission of CDs, and enhanced anti-cancer effect of CB [58]. Moreover, hybrid liposomes have attracted wide interest with numerous applications in controlled release and synergistic treatment of loaded cargos. Li et al. reported liposomes coated mesoporous CNTs with resveratrol (RES) loading for targeted and NIR-triggered chemo/photothermal synergistic cancer therapy [59]. RES is absorbed on the mesoporous CNTs via hydrogen bonding and hydrophobicity. This system can improve the water solubility, stability, and bioavailability of RES, thus improving the anti-tumor therapeutic effect. Simultaneously, the mesoporous CNTs have high-efficiency photothermal conversion ability. Upon NIR irradiation, on the one hand, the local temperature of the tumor site can be increased, thereby directly killing the tumor cells to achieve PTT. On the other hand, light-induced warming is used as a stimulus signal to control the targeted release of drugs at the tumor site. In addition, high temperature can also increase the sensitivity of tumor tissues to chemotherapeutic drugs and achieve the superimposition of the effects of chemotherapy and photothermal effects, so as to achieve synergistic treatment of tumors [59,60].

### 2.2. Liposomes Loading Gold-Based Nanomaterials

As one of the least active metals, gold exhibits high variability in shape, size, and surface chemistry, and has high resistance to degradation and oxidation [61]. Additionally, several types of nanostructures with gold cores have been synthesized for cancer treatment, such as nanoparticles (AuNPs), nanorods (AuNRs), nanocages (AuNCs), nanostars (AuNSs), nanoshells (GNS), and nanoclusters (GNC). Among these, AuNPs have been developed as photothermal conversion sensors in several fields [28]. Interestingly, the photothermal conversion efficiency of gold-based nanomaterials greatly depends on their shape and size. For instance, Jiang et al. reported that the photothermal conversion efficiency of AuNPs is increased from 65.0 ± 1.2% to 80.3 ± 0.8% as the diameter decreased from 50.09 ± 2.34 to 4.98 ± 0.59 nm [62]. Meanwhile, gold-based nanomaterials have obvious surface effects, ultra-small sizes, macroscopic quantum tunneling effects, and local surface plasmon resonance (LSPR) [63]. In the hybrid nanomedicine delivery system, gold-based nanomaterials are able to trigger drug release by strongly absorbing NIR light at the wavelength of 700–900 nm to generate localized hyperthermia [64]. Specifically, this strong and tunable linear NIR absorption can maximally penetrate into the tissue [65]. For a long time, gold-based nanomaterials have been used to treat human rheumatoid arthritis [66]. The biosafety of AuNPs is widely accepted, although they may not fully degrade in the body. Therefore, all of these particular properties have caused gold-based nanomaterials to become the most commonly used photothermal agents. NIR-mediated photothermal trigger is unique because the trigger itself has a therapeutic effect [67]. However, these approaches showed limited drug encapsulation into the inner area of the gold-based nanomaterials. Pure AuNPs show excellent stability, but with a lower LSPR extinction coefficient [68]. After coating with gold shells, the hybrid liposome system exhibited a controllable LSPR absorption peak [68]. Therefore, combining gold-based nanomaterials with liposomes to develop light-addressable carriers as a general platform to realize chem-photothermal therapy became a research hotspot. Jia et al. reported that a positively charged lipid bilayer membrane coated on reduced graphene oxide@gold nanostar (rGO@AuNS-Lipid) for photoacoustic/photothermal dual-modal imaging-guided gene/photothermal synergistic therapy of pancreatic cancer (Figure 2B) [37]. Under 808 nm irradiation, the temperature change (ΔT) is elevated from 13 °C to 41.3 °C as the laser power increased from 0.1 to 0.7 W/cm^2^. Interestingly, ΔT is also affected by the concentration of rGO@AuNS-Lipid. The next laser on-off cycles (six times) test of rGO@AuNS-Lipid show good photothermal stability. Further, after 10 min irradiation (808 nm, 0.1 W/cm^2^), the final temperature of cells in a 96-well plate can arrive at 52.6 °C, proving the excellent photothermal property of GO@AuNS-Lipid. Furthermore, Kwon et al. constructed GNC-labeled thermosensitive liposomes that trigger drug release at the tumor microenvironment under external NIR irradiation [64]. Thermosensitive liposomes containing AuNPs can be triggered by heat to produce hyperthermia, thereby destroying the lipid bilayers and releasing the active components. The co-delivery of DOX and AuNPs based on thermosensitive liposomes involves the synergistic application of diagnostic and therapeutic functions within the same delivery system. DOX has a characteristic fluorescence at 595 nm with an excitation wavelength of 488 nm [69]. The fluorescence signal is quenched and the surface-enhanced Raman scattering (SERS) signal is enhanced upon encapsulating DOX molecules into this hybrid system, which can be used to evaluate the in vivo distribution of drug molecules [70]. Meanwhile, the enhanced SERS signal is helpful for targeted drug delivery [68]. Particularly, the thermosensitive liposomes improve the stability of AuNPs and the loading efficiency of DOX in the blood circulation, as well as the heat-responsive drug release, which ensures the improvement of the anti-tumor efficacy [71].

### 2.3. Liposomes Loading Semiconductor Quantum Dots

Semiconductor quantum dots (QDs) are small nanocrystals with a size of 3–30 nm, which are composed of alloy nanocrystalline colloid or core-shell semiconductor material made of metal group II–VI (such as CdSe‚ CdS‚ CdTe, ZnSe), III–V (such as InP and InAs) and IV-VI (such as PbSe) [72,73]. Owing to their special photophysical properties such as high fluorescence quantum yield, tunable bandgap, and strong size-dependent photoluminescence, QDs have become the ideal nanomaterial designed for targeted antitumor drug delivery and fluorescence imaging [74]. Depending on the size of the core, QDs absorb a wide range of light and emits red-shifted light up to the NIR window, which permits tissue imaging and PTT with deep penetration due to the reduced tissue auto-fluorescence and absorption [16]. Ortega et. al. reported novel rod-like CdTe@MPA-PDA particles based on polydopamine (PDA) loaded with CdTe quantum dots (QDs) capped with mercaptopropionic acid (MPA), which was evaluated as a potential actuator for PTT [75]. Under mild conditions established for the safe and efficient use of lasers, the temperature that was increased in tumor sites was about 10.2 °C, suggesting a good photothermal conversion efficacy. Therefore, the energy of incident light can be converted into heat through QDs, thus achieving photothermal ablation of tumors under NIR irradiation [76]. However, most QDs with high luminous efficiency are insoluble, thus limiting their clinical application [77]. Besides, the potential toxicity of heavy metal ions released from QDs and in vivo nonspecific distribution are still a matter of concern [78]. To this end, surface modification for QDs may be a viable pathway to improve water solubility and bioavailability. However, evidence suggests that some ligands such as peptides, antibodies, and polymers directly modified on the surface of QDs may cause fluorescence quenching and reduce photostability [79,80]. It requires higher doses of QDs to realize ideal therapeutic effects during in vivo applications and thus increasing potential toxicity [34]. Encapsulating QDs in liposomes can protect QDs from oxidation by restricting the oxygen transport to the surface, thereby reducing the inherent cytotoxicity of QDs [81]. Additionally, the use of this hybrid system may allow clinicians to monitor the progress and efficacy of treatment throughout the course of treatment [65]. Currently, numerous QDs such as CdSe, CdSe/ZnS, CdSe/CdZnS, and CdTe have been embedded into the hydrophobic bilayer of liposomes without obviously destroying the integrity of membrane structure [74,82,83]. For instance, in 2016, Lin et al. developed a system of liposomes loaded with CdSeTe/ZnS QDs for brain tumor imaging studies, with a diameter of 175 nm and encapsulation efficiency of 96% [84]. After encapsulating into liposomes, the cellular uptake of QDs was obviously increased compared with pure QDs, reflecting in a higher fluorescence by flow cytometry detection [84]. Samadikhah et. al. also observed increased cellular internalization and lower cytotoxicity of the CdSe-ZnS QD-liposome system [85]. Moreover, researchers investigated the photostability of QDs when changing pH values and observed the protective effect of liposomes on QDs fluorescence [86]. Aizik et al. found that liposomes were able to protect CdSe/CdZnS QDs fluorescence under various conditions, such as low pH, serum-containing medium, and high concentrations of calcium ions [87]. The uptake of free QDs is occurred in a non-specific manner through pinocytosis, while CdSe/CdZnS QD-liposome is internalized in a receptor-mediated process, which is linked to the phagocytosis of monocytes. Further, in vivo animal experiments also proved that monocytes preferentially capture the liposomal delivery system of QDs rather than free QDs [87]. However, liposomes encapsulating QDs may also increase the difficulty of cell internalization because of larger size, thus blocking the release of QDs or drugs from liposomes near the target site [86].

## 3. Synthesis of NIRN-Lips

### 3.1. Preparation Method and Encapsulation Strategy of NIRN-Lips

Given the growing interest in NIRN-Lips, an accurate understanding of how liposomes interact with NIRNs is quite important. Generally, three strategies are developed for NIRNs attachment, which relies on the property of NIRNs. Among them, hydrophobic NIRNs are inserted within the lipid bilayer, while hydrophilic NIRNs are encapsulated into the aqueous core or conjugated on the outer surface (Figure 3) [11]. As shown in Table 1, different methods have been used to prepare NIRNs that possess stable liposomal structures and high fluorescence intensity, including thin-film hydration, solvent injection, ultrasonication, hydrothermal method, ionic interaction assembly method, and extrusion method [84,88,89,90,91,92]. Unencapsulated NIRNs were removed through several purification technologies such as dialysis, ultrafiltration, and ultracentrifugation [11].

In comparison with other methods, thin-film hydration is the most conventionally used method for the preparation of NIRNs because of higher encapsulation efficiency and more stable structure [34]. It is a fabricating process that forms a thin film of single lipid or mixed lipids and is subsequently hydrated by an aqueous solution of NIRNs [93]. This method can achieve three encapsulation strategies for preparing different NIRN-Lips. For hydrophobic NIRNs such as CNHs and QDs, they are more likely to be encapsulated within the lipid bilayer of liposomes during the hydration process. Pippa et al. designed a hybrid liposome system in that CNHs were inserted within the lipid bilayer by using thin-film hydration, with a size range of 80–100 nm [94]. Mengran Xu et. al. fabricated an NIR-controlled nanoplatform using thin-film hydration (WS2QDs-Van@lipo), which was based on the encapsulation of tungsten sulfide QDs (WS2QDs) and antibiotic vancomycin in thermosensitive liposomes (Figure 4A) [95]. Because of the photothermal effect of WS2QDs, the thermosensitive liposomes are disrupted under NIR irradiation to achieve precise drug release, thereby reducing the required dosage concentration of an antibiotic and positioning the drug in the area containing the targeted pathogen [95]. Furthermore, another liposome with a QDs-apomorphine (APO) hybrid was developed, forming spherical vesicles with an average diameter of 142 nm [96]. QDs and APO were mainly located in the lipid membrane and inner core of the liposomes, respectively. Among them, almost 100% QDs and nearly 80% APO were encapsulated. Greater fluorescence intensity was observed in the brains of mice treated with QDs-liposomes compared with free QDs, which was further confirmed by in vivo organs imaging [96]. Compared with organic carriers that self-assemble through hydrophilic and hydrophobic interactions in solution, drug molecules or their derivatives usually conjugate on the NIRNs surface through covalent linkages to acquire higher in vivo stability. For instance, the PEG400-PTX derivative was covalently modified on the surface of AuNPs, the conjugation of which was subsequently encapsulated into liposomes through the thin-film hydration process. The mixed liposomes solved the problems of solubility and stability of AuNPs and also showed high drug loading capacity [97]. In this case, AuNPs were embedded in the phospholipid bilayer along with hydrophobic PTX [97]. However, as summarized in Table 1, most gold-based nanomaterials can be encapsulated into the aqueous core or on the outer surface. The hybrids that AuNRs modified on the surface of liposomes have been successfully synthesized through thin-film hydration, with high structural stability and excellent optical performance [98]. Dynamic light scattering (DLS) results showed that the hybrids exhibit smaller vesicles with average sizes between 100 and 120 nm, while pure liposomes had wider and larger sizes over 200 nm. This phenomenon is closely related to the vesicle stabilization effect of nanorods self-assembly around lipid bilayer, further verified by the decreased zeta potential [98]. The NIR-responsive liposomes were established by incorporating AuNSs with a diameter of 45–50 nm into the aqueous core of liposomes, which allows the conversion of NIR-light energy to heat energy to control the drug release [99]. Another report developed a PTX-TSL-AuNS co-delivery system, which has the advantages of rapid PTX release and easy intracellular uptake under hyperthermia [100]. The encapsulation efficiency and drug loading of PTX are about 92.3% and 7.2%, respectively, suggesting a feasible method for PTX delivery. This co-delivery system exhibits high stability for minimal changes in size and PDI during three weeks at 4 °C [100]. Studies have found that the binding of AuNPs on liposomes’ surface easily generated liposomes aggregation and drug leakage [101]. If AuNPs are inserted into the phospholipid bilayer, micelles may be generated due to AuNPs size exceeding 6 nm [102]. The above problems could be solved when AuNPs appeared in the inner cavity of liposomes [101,102]. Shanshan Xing et. al. successfully designed and prepared an Au/DOX-Lip system that co-encapsulates AuNPs and DOX into its vesicle, which has excellent photothermal conversion efficiency under 808 nm irradiation. AuNPs in the inner core of liposomes were clearly observed in TEM images [101]. For hydrophilic NIRNs such as CDs, to keep stability and fluorescence intensity, embedding them in the inner core of liposomes is the main strategy during the hydration process. Guan et al. reported glycosylated liposomes loading CDs prepared by thin-film hydration [55]. TEM and AFM images showed that the morphology of CDs is relatively uniformly sphere, with a size range of 4–8 nm and an average range of 5–6 nm. Compared with blank liposomes, the size of CDs-Lips does not change while there is a certain difference in morphology. The bumps on the surface of the liposomes are considered to be the result of encapsulating CDs. Moreover, fluorescence intensity and stability of CDs in liposomes were obviously increased in HepG2 cells [55]. Further, our previous work confirmed the photoluminescence emission intensity of negatively charged CDs was strongly enhanced up to 5-fold both in vitro and in vivo after encapsulating into liposomes (Figure 2A) [58]. Interestingly, photoluminescence of the positively charged CDs could also be enhanced 4-fold [58]. In-depth research found that the quantum yield of CDs increased by six times, and the photoluminescence lifetime decreased from 1.03 ns to 0.80 ns. Since the surface of CDs contains a large number of functional groups such as amino, hydroxyl, and carboxyl groups, the photoluminescence performance of CDs can be strongly affected by the interaction with polymer nanoparticles [103]. Factors such as free-radical surface interactions, intramolecular rotational motion, and other non-radiative decay have been reported to impair the photoelectric properties of CDs, resulting in severe photoluminescence inefficiencies [104,105]. Therefore, embedding CDs in nanoscale liposomes may effectively reduce the vibration and rotation of CDs, reduce the surface interaction of free radicals, and increase the decay process, thus improving the luminescence intensity of CDs compared with pure CDs.

Seed-mediated method is considered subordinate to the thin-film hydration approach, which is unique for preparing gold-based nanomaterials [106]. Preparing an AuNP/liposome hybrid system in this way requires two steps. Firstly, blank liposomes or drug-loading liposomes were fabricated during thin-film hydration. Secondly, GNS with strong photothermal conversion ability were grown on the surface of liposomes by the seed-mediated method. During the GNS synthesis process, the gold seed forms a nucleation center and additional AuNPs subsequently deposit on its surface [107,108]. For example, a multifunctional responsive drug carrier (GNS@CTS@RES-Lips) was prepared by loading RES into chitosan (CTS) modified liposomes, and coated with GNS [107]. A growing solution is composed of auric chloride, potassium carbonate, and ascorbic acid. Gold seed was provided with sodium borohydride and then connected with CTS@RES-Lips by electrostatic action. After that, hydroxyl ion is provided to auric chloride to form a stable gold hydroxide coordination compound. Under this condition, the self-nucleation phenomenon of gold is suppressed, and the gold core becomes larger to form GNS. Finally, gold ion is further reduced with ascorbic acid to further form a continuous GNS layer [107]. Based on the same method, Yanan Li et al. reported an HER2-modified thermosensitive liposome (immunoliposome) auxiliary complex, by reducing the gold nanoclusters on the surface (GTSL-CYC-HER2) to obtain a novel type of bio-plasmonic resonance structure carrier (Figure 4B) [35]. In another study, a biodegradable liposome-AuNP carrier was designed for curcumin (CUR) delivery. CUR is embedded in liposomes with an encapsulation rate of about 70%. The gold coating allows this hybrid system to specifically absorb light energy at 780 nm through the LSPR effect and convert it into heat energy to trigger CUR release. More importantly, CUR in the liposome-AuNP system was unaffected in eight hours, while free CUR or CUR-Lips without AuNPs decoction showed a certain degradation [109]. Over the past decade, some preparing methods such as solvent injection, ultrasonication, and hydrothermal method have also been developed to prepare NIRN-Lips [89,91,110,111]. Nevertheless, in terms of particle size uniformity and structural stability, NIRN-Lips prepared by the thin film dispersion method still have irreplaceable advantages.

### 3.2. Design Consideration and Surface Modification of NIRN-Lips

Encapsulating the NIRNs into liposomes is one of the most promising pathways to overcome limitations such as high hydrophobicity and low bioavailability. However, the size, shape, lipid composition, and surface charge of these nanomaterials have a significant impact on the preparation and application of NIRN-Lips, which is depicted in Figure 3 [72]. Among them, the size of NIRNs should be the first design consideration during the liposome encapsulation process. It is important that AuNPs must be designed in the size range of 2–5 nm to ensure the lipid film integrity although it may still cause considerable deformation [144]. These small AuNPs are conducive to clear from the kidneys and achieve controlled release from liposomes triggered by NIR light [11]. Many studies have confirmed that either naked or targeted AuNPs with diameters less than 20 nm have high small-molecule-like affinity in vitro and in vivo, exhibiting longer tumor retention time and faster clearance of normal tissue [124]. AuNPs may penetrate deeper tissues and completely sustain PTT if the size of AuNPs is smaller than the intercellular space of cancer cells [101]. Moreover, AuNRs with a length of 30–100 nm and an aspect ratio of less than 5, can indeed strongly absorb light in the 700–1000 nm range (NIR light) through surface plasmon resonance, and convert it into heat through fast electrons-phonon/phonon-phonon processing, which is a useful candidate for cancer PTT [145]. Depending on the different sizes, AuNPs exhibit a local surface plasmon peak, which is often used for the SERS [146,147]. Faried et al. designed SERS-based lipid membrane system characterization and further compared the effect of AuNPs size on SERS performance [148]. The results indicated that AuNPs with various particle sizes were encapsulated in different sites by liposomes, showing a difference in optical properties and thereby affecting the practical application. Except for AuNPs, the insertion of the hydrophobic QDs into the lipid bilayer is also limited by their particle size. Reports have shown that only QDs with a size less than 5 nm can be easily embedded in the hydrophobic part of the bilayer [149]. For QDs larger than 6 nm, it is observed that QDs were embedded in the hydrophobic micelles formed by phospholipid [150]. Wlodek et al. reported incorporation of hydrophobic cadmium sulphide quantum dots (CdS QDs) into mixed 1-palmitoyl-2-oleoyl-sn-glycero-3-phosphoethanolamine (POPE)/1-palmitoyl-2-oleoyl-sn-glycero-3-phosphocholine (POPC) liposomes. The QDs were embedded in the hydrophobic regions of the liposomes and the supported bilayers, which retained the QDs fluorescent properties. Particularly, the damage degree of QDs to the supporting bilayer structure is related to the size of QDs. CdS QDs with sizes of 3.8 nm and 4.9 nm were optimal to be embedded within membrane bilayer [74]. Carbon-based nanomaterials, especially CDs with a small diameter of about 10 nm, are easily filtered by the liver and kidney [151]. To escape this filtration, Geng et al. increased the size by electrostatically attaching CDs to tungsten disulphide nanosheets to enhance toxicity to cancer cells [152].

The shape of NIRNs also displays an important impact. In comparison with QDs, gold-based nanomaterials have various shapes such as rodlike, spherelike, and starlike, which have significant effects on the in vitro and in vivo behaviors. For example, the absorption of spherical AuNPs to tumor cells was much greater than that of rod-shaped AuNPs due to the large surface area [125]. In other words, because of the amplification of the EPR effect, spherical AuNPs are easier to specifically accumulate in the tumor tissue [153]. However, it is also reported that rodlike nanoparticles have a longer lifetime in the bloodstream and penetrate tumors more efficiently than spherical particles of matched size owing to their least vascular collisions and the shortest size [154]. Gold-based nanomaterials with sharp or slender ends have higher thermal capacity than spherical AuNPs [155,156]. This phenomenon is due to the enhancement of laser light by the elongated or branched structures, resulting in increased absorption of light and heat generation [53]. The prominent morphology of gold-based nanomaterials enhances their polarizability. At the plasmon resonance wavelength, increasing polarizability leads to a red shift [53]. Moreover, compared with solid AuNPs, hollow AuNPs have stronger photothermal conversion efficacy and more adjustable absorption wavelength in the range of 550 to 900 nm because of their larger cross-section [157]. Li et al. found that liposome-loaded hollow AuNPs exhibit better treatment-related heating capacity and significant thermo-chemotherapy efficiency than solid AuNPs [125]. In particular, the photothermal conversion efficiency of AuNPs and hollow AuNPs is very different. Hollow AuNPs have the advantage of near-infrared surface plasmon tunability, resulting in tumor photothermal ablation with light penetration of 800 nm in tissue. The co-encapsulated DOX is released under NIR response to hyperthermia caused by hollow AuNPs, and the local drug concentration increased with the disintegration of the liposome membrane, thereby producing a synergistic cytotoxicity response [125]. Additionally, the dissimilarity in the shape of carbon-based nanomaterials also affects their properties and applications [94,158]. For example, CNHs do not have high aspect ratio problems associated with the toxicity observed in longer CNTs [94]. The irregular structure of CNHs allows the opening of nanoscale windows at the tip and side walls, resulting in the easier absorption and release of bioactive molecules [94].

As illustrated in Table 1, a large number of lipids can be applied in preparing liposomes, including cationic lipids, zwitterionic lipids, and anionic lipids. Lipid types can significantly affect the properties and applications of liposomes. The surface composition and charge of liposomes are parameters that regulate their interactions with cells and absorption mechanisms. Generally, both neutral and anionic liposomes exhibit weaker binding to cell membranes [86]. However, cationic liposomes composed of positively charged lipids can be nonspecifically linked to cell membranes with a negative charge by electrostatic interactions, thus facilitating the delivery of therapeutic agents into cells [159]. Al-Jamal et al. reported that cationic QD-liposomes with mixed lipids of dioleoyl phosphatidylcholine (DOPC), cholesterol (Chol) and N-[1-(2,3 dioleoyloxy)propyl]-N,N,N-trimethylammonium methyl sulfate (DOTAP) exhibit faster blood clearance than neutral QD-liposomes with mixed lipids of 1,2-distearoyl-sn-glycero-3-phosphoethanolamineN-[methoxy(polyethyleneglycol)-2000] (DSPE-PEG2000), DOPC, and Chol, suggesting a higher intracellular uptake in liver [72,160]. Moreover, they also confirmed the enhanced internalization and cellular binding of cationic QD-liposomes into cancer cells [161]. Pippa et al. prepared a cationic CNH/liposome system that exhibited stronger cytotoxicity, which might be attributed to the positive charge of CNHs affecting their binding and release from liposomes [94].

Conventional liposomes are readily absorbed by the reticuloendothelial system during intravenous administration, resulting in a short circulation time [86]. To increase circulation time, the liposomes modified with PEG have been developed. Luo et al. reported a facile synthesis of RES-loaded folate-terminated PEG-phospholipid coated reduced graphene oxide nano-assembly, exhibiting a higher circulation time compared with pure RES [141]. After designing folic acid (FA) with drugs or their delivery systems, targeting the FA receptor is considered one of the most promising approaches in improving drug uptake [162]. Because of the FA-mediated targeted delivery, FA-PEG-Lip@rGO/RES can deliver RES to MCF-7 cells with high specificity and ideal efficiency. Moreover, the FA-PEG-Lip@rGO delivery system shows no cytotoxicity, suggesting its applicability as a drug carrier [141]. As shown in Table 2, numerous NIRN-Lips modified with FA have been developed to target a variety of cancer cells. Liposomal nanotheranostics loaded with AuNPs and GQDs are functionalized with FA targeting ligands, which are used to deliver DOX. Owing to the targeting effect of FA, hybrid liposomes injected subcutaneously and intravenously showed strong 4T1 breast tumors-binding ability for at least 48 h [122]. Chen et al. developed a negatively charged FA coating pH-sensitive liposome to deliver DOX@CDs, aiming to increase retention and permeability at the tumor sites [119]. In comparison to free DOX@CDs, FA-DOX@CDs-Lips showed higher tumor aggregation in 4T1 tumor-bearing mice and further increased cytotoxicity to tumor cells. Upon arriving at the tumor site, the specific binding of FA to its receptor on the tumor cell surface triggered the endocytosis process. FA-DOX@CDs-Lips was broken up inside the cells, followed by the embedded DOX@CDs escaped from the endosome [119]. Nowadays, liposomes modified with cell-penetrating peptides (CPPs) have been widely concerned with enhancing cell permeability. Even at very low concentrations, CPPs can penetrate cell membranes without using specific receptors, which will not cause obvious membrane damage [163]. A novel CPP polyarginine-conjugated CDs modified liposomes was prepared for transdermal CUR delivery. Compared to conventional liposomes, CPP-CDs-CUR-Lips with nanoscale spherical shapes, have higher stability, longer circulation time, and better cellular compatibility. Both in vitro and in vivo studies revealed that CPP-CDs-Lips are able to enhance deep skin penetration, suggesting an excellent transdermal drug carrier [112]. Transferrin (Tf) receptor is a membrane-associated glycoprotein on the cell surface that regulates iron homeostasis and cell growth [164]. Due to the fast proliferation of cancer cells and the large demand for iron, the expression of the Tf receptor on cancer cells is 100 times higher than the average expression of normal cells [165,166]. Therefore, Tf has been developed as a targeted ligand for nanocarriers delivering therapeutic and diagnostic agents to cancer cells [166]. The preparation of Tf modified gold-based theranostic liposomes has been reported for brain-targeted tumor therapy [89]. In vivo studies showed that the Tf targeted gold liposomes delivered docetaxel (DOC) to the brain at 3.7-, 2.7-, and 4.1-fold higher than the marketed DOC formulation after 30, 120, and 240 min of treatment, respectively. These results confirmed that Tf-modified liposomes can significantly enhance DOC transport via the blood-brain barrier and contribute to imaging and diagnosis. Compared with non-targeted preparations or marketed DOC formulation, the higher percentage of Tf-targeted gold liposomes in the brain may be attributed to higher expression of Tf receptors on the blood–brain barrier, which promotes receptor-mediated endocytosis [89].

## 4. NIR Light-Triggered Drug Release Mechanism

The NIR light-triggered drug release mechanism is considered a photothermal process [168]. Several mechanisms have been proposed as a means of NIR light-triggered membrane instability in liposomes to promote cargo release, such as photoinduced oxidation, photocrosslinking, photoisomerization, photolysis, and photothermal release [169]. However, most NIRN-Lips systems depend on the photothermal effect that is achieved by the conversion of light into heat to induce liposome permeabilization [11]. Therefore, it is necessary to understand the photothermal conversion mechanism of these NIRNs. Upon absorbing electromagnetic radiation, NIRNs have the ability to generate lattice vibration or electronic oscillation and cause the rising temperature, thus achieving photothermal conversion [28]. There are some differences in the photothermal conversion mechanisms for various NIRNs. Generally, it can be categorized into three mechanisms, including the LSPR effect, electron-hole generation and relaxation, and conjugation or hyperconjugation effect [170]. Firstly, based on a previous description, gold-based nanomaterials possess their unique LSPR effects. Upon NIR light irradiation, free electrons existing on the surface of gold-based nanomaterials are excited to produce asymmetrically scattered electron clouds. Further, since the frequency of light is consistent with the LSPR frequency of AuNPs, the plasma nanomaterials absorb abundant light and generate highly amplified local electric fields. Surface plasmon oscillation is attenuated by radiating its energy, causing the absorbed light to be converted into heat [28,171,172]. Both internal and external factors can affect the photothermal conversion efficiency of gold-based nanomaterials, such as shape, size, composition, and pH [173]. Secondly, electron-hole generation and relaxation mainly occur in QDs. The absorption of light by QDs mainly depends on the inherent absorption band gap between its valence band and conduction band, including transition metal nitrides, borides, and sulfides. When irradiated by incident light with energy greater than the band gap of the semiconductor, the transition metal ion will absorb the photon energy, the valence band electron is excited, and then transition to the conduction band, generating electrons and holes to the valence band. The electron-hole pair relaxes to the edge of the energy band and releases additional energy in the form of phonons, which is converted into heat through non-radiative decay [170]. For wide band gap semiconductors, due to most of the absorbed energy re-emitted as photons, electron-hole pair recombination is prone to occur, thereby reducing the photothermal conversion efficiency [28,170,174]. Thirdly, carbon-based nanomaterials realize light-to-heat conversion through conjugation or hyperconjugation effect. In the conjugated system, the overlap of π electrons or the interaction between π bond and p-orbital electrons redistributes the electron density, resulting in the conjugation effect. The interaction between an electron of σ bond and nearby vacant or partially filled p-orbital can also lead to conjugation effects, which is known as hyperconjugation. Both the conjugation effect and hyperconjugation effect allow strong absorption in the NIR range and accelerate electron migration. Electrons in π orbitals are excited and jump further into π* orbitals, releasing heat when they returned to the ground state [170,175].

The photophysical-induced cargo release from liposomes usually relies on photothermal conversion after photons are absorbed from an external light source, which can cause thermal and/or mechanical processes in the bilayer film or its surrounding medium. During this process, no chemical changes occurred in the liposome membrane or the structure related to the liposome membrane [176]. As illustrated in Figure 5, the possible mechanisms of NIR light-triggered drug release are attributed to the photothermal effect produced by NIRNs in NIRN-Lips. Firstly, during NIR radiation, the photothermal effects cause disturbance and increase the permeability of local liposome films near the NIRNs. Secondly, the heat generated by NIRNs under NIR irradiation increases the temperature of the entire liposome due to its longer diffusion and duration. The phase transition of the phospholipid bilayer leads to an increase in the permeability of the entire liposome membrane, triggering the release of the drug [132]. Supported by this concept, Wang et al. confirmed GNS@CTS@RES-Lips enables on-demand drug release to minimize drug toxicity to normal cells [107]. The release of RES was significantly increased after 5 min irradiation with NIR light. After turning off the NIR light, the drug release rate returned to the normal rate. The obvious increase in drug release rate under NIR light is attributed to the local hyperthermia generated by the effective photothermal conversion of GNS, which can effectively induce the phase transformation of liposomes from a gel state to a liquid crystal state upon reaching the critical solution temperature. In the process, the hydrophobic tail of the phospholipid begins to rotate and wiggle, thereby triggering drug release [177,178]. Nevertheless, the drug release mainly exhibits a slow diffusion process below the critical temperature due to the vertical state of the hydrophobic tail of phospholipid [107]. Furthermore, You et al. also demonstrated the main contribution of DOX and HAuNS-TSL triggering the release of DOX is the heat diffusion leading to phase transition and increased permeability of the entire liposome membrane. Moreover, the enhanced NIR light intensity causes the increased DOX release [132]. As for carbon-based nanomaterials, there are similar NIR light-triggered drug release mechanisms as gold-based nanomaterials. Liposomes loading CDs, CNTs, and GO showed sustained drug release under NIR irradiation [58,118,179]. For example, GO produces a strong photothermal effect that raises the temperature beyond the melting temperature of the phospholipid molecules, leading to disordered lipid bilayer arrangement, followed by enhanced lipid membrane permeability and cargo leakage [176]. Another report also found that the DOX release of graphene nanosheets-DOX-Lips was enhanced than that of DOX-Lips after being exposed to NIR light, which is attributed to the effect of graphene nanosheets on the structural stability of liposomes [121]. DOX release from the hybrid system of QDs and liposomes exhibits rapid release in the first 5 min and sustained release afterwards. Moreover, the release rate can be adjusted by the laser power. TEM images indicated that most liposomes remained intact and some vesicle fragments were observed after irradiation with 808 nm laser at different power densities for 30 min. QDs within the bilayer of liposomes convert light to heat energy, thereby increasing the fluidity of the lipid membrane and even destroying liposome structure, resulting in cargo release from liposomes [116].

## 5. Application

### 5.1. Cell Death Mechanism Induced by PTT

At present, there are still some ambiguities about the specific mechanisms of cell death after PTT while this strategy can effectively kill cells and ablate tumors [25]. A better understanding of the cell death pathway induced by PTT is needed for designing more appropriate and effective treatment plans. Numerous factors are linked to this process, including tumor types, NIR light irradiation conditions, internalization degree of NIRNs, and heat absorbed by cells [180]. The main mechanism of PTT is that local high temperature induces various changes in tumor cells. For example, hyperthermia induced by PTT triggers dying tumor cells to release antigens, pro-inflammatory cytokines, and immunogenic intracellular substrates, thereby promoting immune activation [25]. Currently, researchers have used an infrared thermal imaging camera (LaserSight, Optris, Germany) to visualize the rising temperature of NIRNs-Lips [88,118]. As shown in Table 3, upon NIR irradiation, the temperature of all the recently developed NIRN-Lips can reach 43–66 °C within 5–20 min, suggesting a good photothermal conversion efficiency. According to previous reports, a temperature of 43 °C is sufficient for hyperthermia, which usually results in the denaturation and destruction of protein membranes and ultimately cancer cell death [181]. Under high temperatures, normal cells can be in the heat stress state [182]. During this process, the expression of heat shock protein is upregulated, helping to reduce the damage caused by protein denaturation and inhibit the activation of apoptosis-related pathways [183]. PTT mediated by gold-based nanomaterials can induce physiological and biological changes in the surrounding tumor environment, which can maximize the cytotoxicity of PTT and improve the efficacy of secondary therapy. Interestingly, PTT can enhance the delivery of AuNPs or secondary therapeutic molecules by amplifying the leakage of tumor blood vessels and the permeability of the extracellular matrix, thereby overcoming the penetration and retention limitations of tumors [173]. Riley et al. also concluded the mechanisms of cell death triggered by PTT [173]. Laser energy intensity has a significant effect on PTT. For example, traditional PTT under high-energy irradiation causes rapid NIRNs heating, resulting in cell necrosis, which is effective for the ablation of the established tumors [184]. However, this strategy triggers the inflammation induced by cellular waste release, causing secondary tumor growth. Conversely, PTT with low energy irradiation induces apoptosis to inhibit inflammation by producing TGF-β and other anti-inflammatory molecules, bringing a beneficial immunogenic response. In this process, photothermal activity and concentration in tumor tissue of NIRNs, and light power have a significant influence on PTT efficacy [185]. Further, excessive reactive oxygen species (ROS) are generated by PTT under NIR irradiation, including hydroxyl radicals, hydrogen peroxide superoxide, and singlet oxygen [186]. Overproduced ROS can damage DNA, proteins, and lipids, and eventually lead to cancer cells apoptosis, which is considered to be another anti-cancer mechanism of PTT [187]. For example, Prasad et al. reported that CDs convert NIR light to heat and generate ROS, confirming photo-triggering 4T1 cell death and breast tumor regression [110].

### 5.2. NIRN-Lips Mediated Cancer PTT Treatment

NIRN-Lips mediated PTT is a novel type of nanotechnology that converts light energy to heat energy to kill cancer cells [173]. As shown in Table 3 and Figure 6, we provide a brief summary of the recently developed NIRN-Lips for cancer PTT. For example, liposomes loaded with NIR photoactive functional red fluorescent CDs extracted from mango leaves have been shown to be useful for localized tumor imaging and NIR light-mediated tumor growth inhibition. After 5 min of continuous NIR exposure (808 nm, 1 W), the temperature at the tumor site is increased to 62 °C, showing a good thermal response due to the unsaturated CDs [188]. Further, the designed hybrid liposomes collapsed into small particles less than 50 nm with uniform distribution in the deep tumor environment, enhancing the effective heat distribution of the entire tumor environment [110]. One of the main problems of PTT is that cell death mainly occurs through necrosis under high temperatures over 50 °C. Necrosis is a pro-inflammatory signal that is a threat to the cell environment, however, cell death caused by apoptosis is beneficial to cancer treatment [189]. Small molecules such as quercetin can be combined with PTT to induce cell death via apoptosis [190]. Pradhan et al. designed quercetin-loaded gold-coated liposomes that can specifically induce Huh-7 cells apoptosis after PTT. Upon 750 nm light irradiation, QE-LipsAu shows a similar temperature rise compared with LipsAu. QE-LipsAu with a photothermal conversion efficiency of 75% shows the increased PTT efficacy over LipsAu [137].

However, owing to the NIR light scattering, it is difficult to completely achieve tumor ablation by PTT alone [107]. PTT combined with chemotherapy, namely, chemo-photothermal therapy, has been used to realize the controlled release of chemotherapeutic agents under a light stimulus. As previously mentioned, hyperthermia increases the permeability of blood vessels and cell membranes. In comparison with the above therapy alone, chemo-photothermal therapy exhibits better therapeutic efficacy due to the enhanced drug contents under the rising temperature at the tumor site [191]. Furthermore, the cytotoxicity of the drug may be enhanced by heating. Photoresponsive-induced hyperthermia can compensate for the decreased therapeutic effect caused by tumor drug resistance [192]. NIRNs-mediated PTT provides local, tumor-specific heating that avoids the harmful effects of widespread high temperatures [173]. For example, the thermo-pH dual responsive liposomes were developed by loading RES in CTS modified liposomes and coated with GNS, with broad NIR absorbance and high photothermal conversion ability. GNS@CTS@RES-Lips under NIR irradiation displayed a stronger therapeutic effect on HeLa cells compared with the single chemotherapy and PTT, which is attributed to the enhanced intracellular uptake and on-demand pH/photothermal-sensitive release of RES [107]. Further, an FA-terminated PEG-phospholipid coated reduced GO nano-assembly was designed to protect RES from UV-induced instability. Under 780 nm laser irradiation, FA-PEG-Lip@rGO/Res exhibited a highly-effective combination of chemotherapy and PTT to eradicate xenograft tumors through a single-dose intra-tumoral injection [141]. Recently, except for heat, ROS has been considered as a side product of PTT [122]. Thus, the combined effects of heat and ROS generated from PTT show oxidative and thermal damage of cancer cells. Prasad et al. reported FA-modified liposomal nanotheranostics loaded with AuNPs, GQDs, and DOX. Due to the heat and ROS produced, targeting FA-GQDs/AuNPs-Lips nanohybrids demonstrated tumor reduction mediated by 750 nm light irradiation [122]. Compared to conventional liposomes, thermosensitive liposomes with low cytotoxicity and high stability demonstrated the hyperthermia-triggered release. DOX and GNPs or HGNPs were loaded into thermosensitive liposomes, in which GNPs and HGNPs acted as “nanoswitches” to directly kill tumor cells through hyperthermia and triggered the rapid release of DOX from intratumor thermosensitive liposomes under 808 nm light irradiation. This co-delivery of HGNPs and DOX-TSL exhibits a synergistic cytotoxic response, resulting in an 8-fold increase in anticancer efficacy and prolonged survival compared to GNPs and DOX-TLS [125]. Moreover, another report developed a novel NIR light-sensitive liposome consisting of HAuNS and DOX [132]. In comparison to the treatment of single DOX-TSL and HAuNS-TSL, the DOX and HAuNS-TSL group exhibits greater cytotoxicity and stronger antitumor efficacy under 808 nm laser irradiation [132]. It may be attributed to the increased intracellular DOX concentration because of the triggered release of DOX from hybrid liposomes. Further, after NIR irradiation, there is a synergistic interaction between the photothermal effect and DOX cytotoxic effect. In another study, DOC and GNRs were loaded into the liposomes as chemotherapy and thermotherapy agents. They also found that photothermal therapy enhances anticancer efficacy by increasing DOC release and intracellular delivery to PC-3 cells [127].

Currently, gene therapy is a therapeutic strategy to correct or compensate fundamentally for basic defects and abnormal genes [193]. Particularly, small-interfering RNAs (siRNAs) have received great attention due to their ability to regulate protein expression by silencing mRNA [194]. In the process of combining with PTT, NIRNs are considered as gene delivery carriers, which can increase serum stability of siRNAs and control siRNAs release [195]. Meanwhile, NIRNs binding genes can effectively target to the tumor site, overcoming the limitation of incomplete elimination of cancer cells after PTT alone [196]. For example, Zhao et al. fabricated the temperature-sensitive CNT-PS/siRNA nanoparticle for synergistic PTT and gene therapy for cancer cells (Figure 7A) [36]. Upon NIR irritation, CNT/siRNA inhibits tumor growth by silencing the expression of survivin while exhibiting photothermal effects. Due to peptide lipid (PL) and sucrose laurate (SL) coating, it is very effective for systemic delivery to tumor sites and for promoting siRNA release due to temperature-sensitive lipid phase transitions. Further, Jia et al. reported a positively charged lipid bilayer membrane is coated on reduced graphene oxide@gold nanostar (rGADA) for photoacoustic/photothermal dual-modal imaging-guided gene/photothermal synergistic therapy of pancreatic cancer (Figure 7B) [37]. Photothermal and gene (targeting G12V mutant K-Ras gene) synergistic treatment showed excellent anti-cancer effects and anti-liver metastasis on pancreatic cancer tumor-bearing mice.

## 6. Challenges and Futures of NIRN-Lips

There is an emerging demand for NIRN-Lips systems that permit controlled and triggered release upon light stimulation, which will ultimately lead to the precise treatment of the tumor. However, to make the above-mentioned system clinically available, certain bottlenecks need to be overcome.

The first and foremost barrier in the way of successful NIRN-Lips is the long-term degradation mechanism and potential toxicity of these NIRNs in vivo. Ideally, nanostructures should be completely removed from the body within a certain period of time. In fact, most NIRNs larger than 8 nm in size are difficult to be cleared through the liver and kidneys [197,198]. Particles smaller than 8 nm can only be minimally modified on the surface to improve circulation time and tumor targeting, which is easy to be removed by the mononuclear phagocyte system before accumulating at the tumor site [88]. As we discussed previously, NIRNs with suitable sizes should be fabricated to ensure minimal toxicity and maximal efficacy. Many in vitro and in vivo studies on animal toxicity have been conducted. However, studies on human toxicity are very limited. Future clinical research should be well designed. Secondly, some issues on the structural stability of NIRN-Lips should be addressed. Almost all liposomes are at risk of leakage once they enter the body. It remains unclear whether the incorporation of NIRNs will reduce the stability of liposomes and thereby making them leakier, especially being embedded in lipid bilayers and coated on the surface. Consequently, the optimized formulation is of great significance to protect NIRNs and drug release [72]. In this process, microfluidic technology can be used to establish a cardiopulmonary bypass model to test the stability of NIRN-Lips under high shear stress [11]. Thirdly, the NIR light-responsive liposomal platform is positioned as a drug carrier for skin tissues. Despite a lot of encouraging advances, the tissue penetration of laser light is still considered a limitation to cancer treatment in clinical [199]. Since, high power laser cause collateral damage to the normal tissue surrounding the tumor; whereas low power laser may lead to tumor recurrence, due to inadequate tumor accumulation or thermal resistance of tumors induced by heat shock protein and hypoxia microenvironment of the tumor, leading to tumor metastasis. In addition, the tissue penetration of laser light using animal models is scarcely applicable to humans since in animals, but not in humans, the penetration of the laser light is sufficient to induce photochemical reactions [200].

Conventional PTT as a single treatment is challenging since increased inflammation and secondary tumor growth are caused by cell necrosis. It is difficult to ensure complete ablation of all tumor cells, and PTT is not feasible as a treatment for metastatic disease. Due to the clinical application of PTT is often used as an adjuvant therapy, combining PTT with chemotherapy may provide a synergistic effect, which can improve the feasibility of tumor cure and prevent tumor metastasis and recurrence [201]. With these goals in mind, multidisciplinary cooperation is required to bring significant opportunities for the development of valuable tumor treatment in near future, including pharmaceutics, polymer material science, bioengineering, and clinical medicine.

## 7. Conclusions

Overall, we have elaborated on the recent advances in liposomes loading a large number of NIRNs for PTT. These NIRNs possess many unique properties such as high fluorescence quantum yield, tunable bandgap, and strong size-dependent photoluminescence, providing an opportunity to achieve controllable light-triggered drug release. During the fabricating process, some factors influencing the therapeutic effect of PTT should be considered carefully, including the method of NIRNs attachment, size, shape, and surface charge of NIRNs, and lipid composition. NIRN-Lips with surface-modified targeting ligands have been designed to maximize the tumor accumulation of therapeutic agents. In addition, we discussed the photothermal conversion mechanism of NIRNs that is closely associated with the cargo release from NIRN-Lips, helping to fully understand the in vitro and in vivo behaviors. Furthermore, a brief summary of the recently developed NIRN-Lips for cancer PTT is provided. PTT combined with chemotherapy or gene therapy shows synergistically enhanced therapeutic outcomes without additional toxicity. Therefore, the most important aspect of future research is the need to address the possible human clinical studies to confirm the application of NIRN-Lips in the treatment of various types of cancer.

## Figures and Tables

**Figure 1 pharmaceutics-13-02070-f001:**
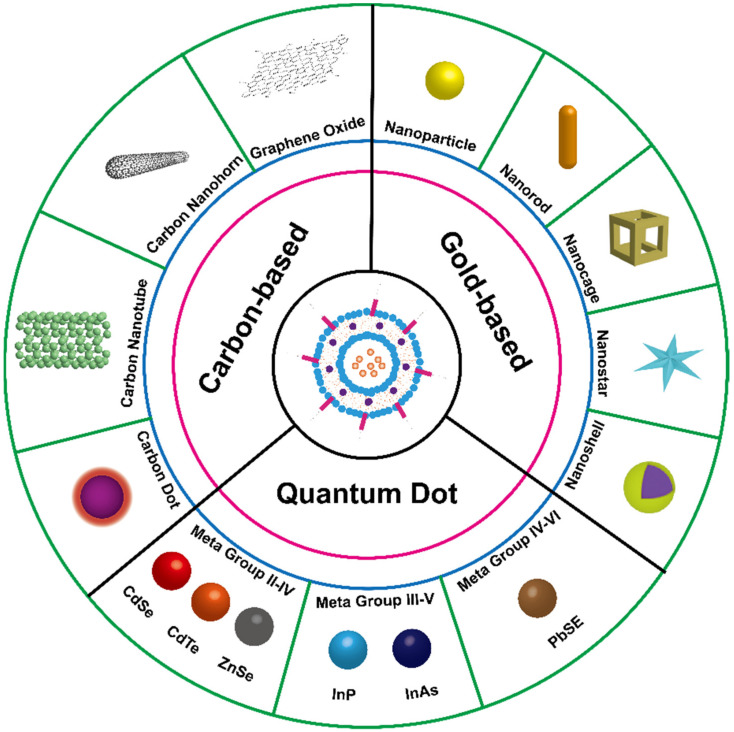
Schematic illustration of liposomes loading various NIRNs for cancer PTT.

**Figure 2 pharmaceutics-13-02070-f002:**
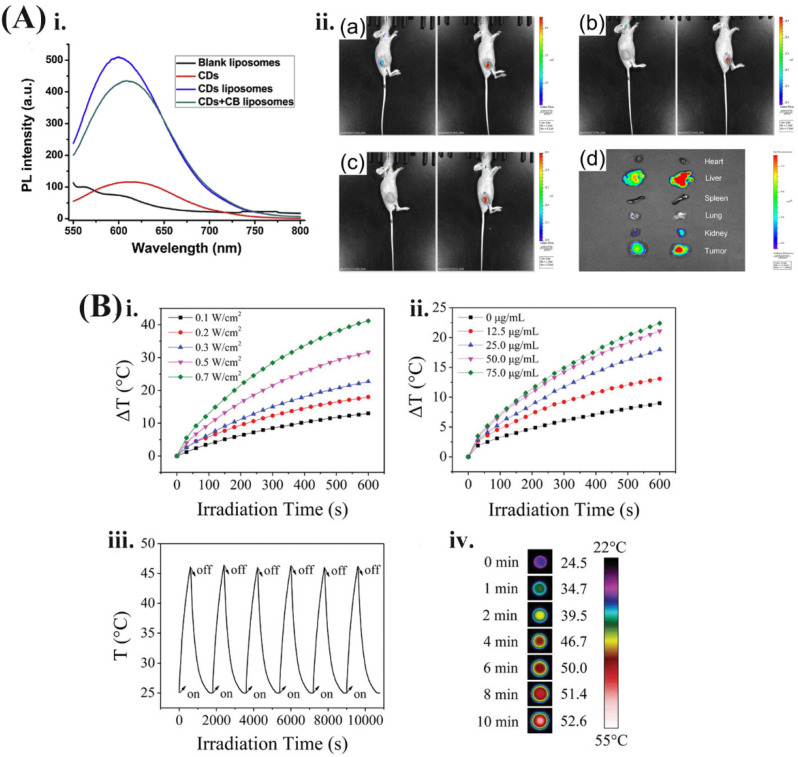
(**A**). (**i**) Photoluminescence intensity (PL) spectra of CDs, blank liposomes, CDs liposomes, and CDs + CB liposomes at excitation of 500 nm. (**ii**) (**a**–**c**) In vivo PL images of nude mice treated with intratumor injection of CDs (the left) and CDs + CB liposomes (the right) 0.25 h, 1 h, and 2 h post injection, respectively; (**d**) ex vivo images of mice tissues 24 h post tail intravenous injection of CDs (the left) and CDs + CB liposomes (the right). The color bars represent the PL intensity. Reprinted with permission from [58], Elsevier, 2018. (**B**) Measurement of photothermal property and photostability of rGO@AuNS-Lipid. (**i**) Temperature changes (ΔT) of the solution containing 60 µg/mL rGO@AuNS-Lipid irradiated at various laser power densities. (**ii**) Temperature change of the solution containing different concentrations of rGO@AuNS-Lipid irradiated at a laser power density of 0.3 W/cm^2^. (**iii**) Thermal stability of rGO@AuNS-Lipid (continuous six cycles of laser on/off) under 808 nm laser of 0.3 W/cm^2^. (**iv**) Infrared thermographic images recorded with a thermal infrared camera of rGO@AuNS-Lipid in 96-well plate irradiated by 808 nm laser at a power intensity of 0.1 W/cm^2^. Reprinted with permission from [37], John Wiley and Sons, 2020.

**Figure 3 pharmaceutics-13-02070-f003:**
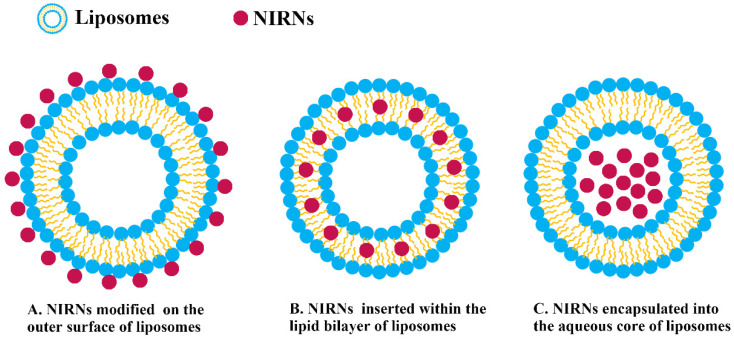
Three strategies developed for NIRNs attaching to liposomes.

**Figure 4 pharmaceutics-13-02070-f004:**
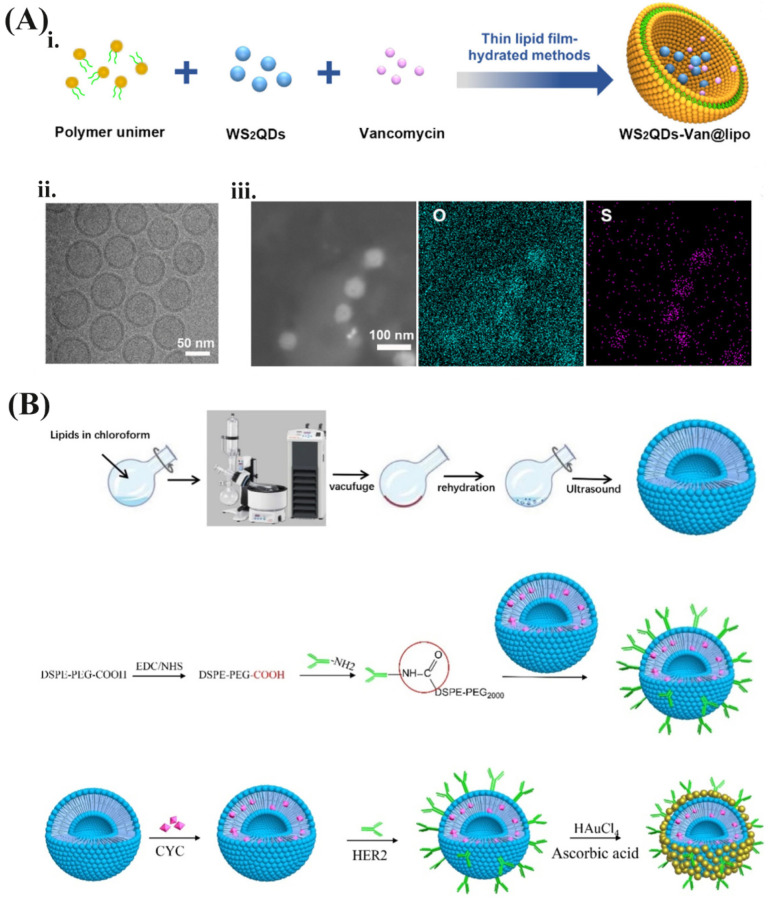
(**A**). (**i**) Schematic illustration of the preparation process of WS2QDs-Van@lipo. (**ii**) Cryo-electron microscopy images of WS2QDs-Van@lipo. Scale bar: 50 nm. (**iii**) Elemental mapping images (O in blue, S in purple) of the WS2QDs-Van@lipo obtained with TEM. Scale bar: 100 nm. Reprinted with permission from [95], American Chemical Society, 2020. (**B**) Illustration of dual-targeting GTSL-CYC-HER2 and its intracellular trafficking pathway in tumor cells. Reprinted with permission from [35], Springer Nature, 2021.

**Figure 5 pharmaceutics-13-02070-f005:**
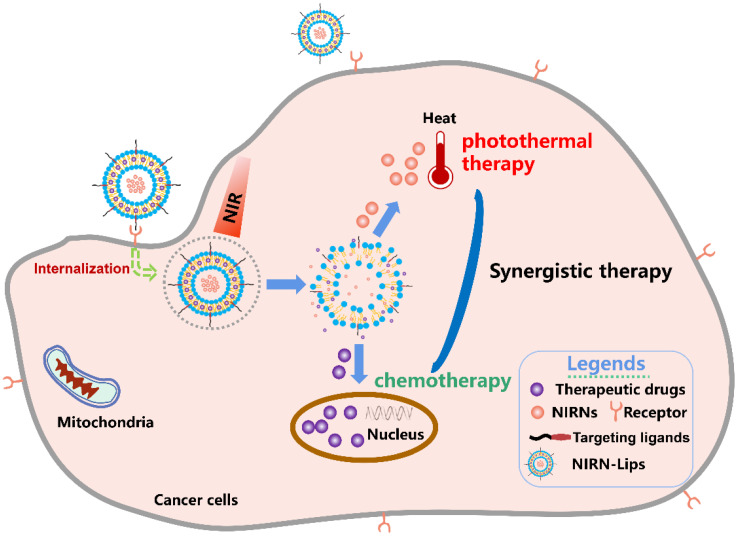
NIR light-triggered drug release mechanism of NIRN-Lips.

**Figure 6 pharmaceutics-13-02070-f006:**
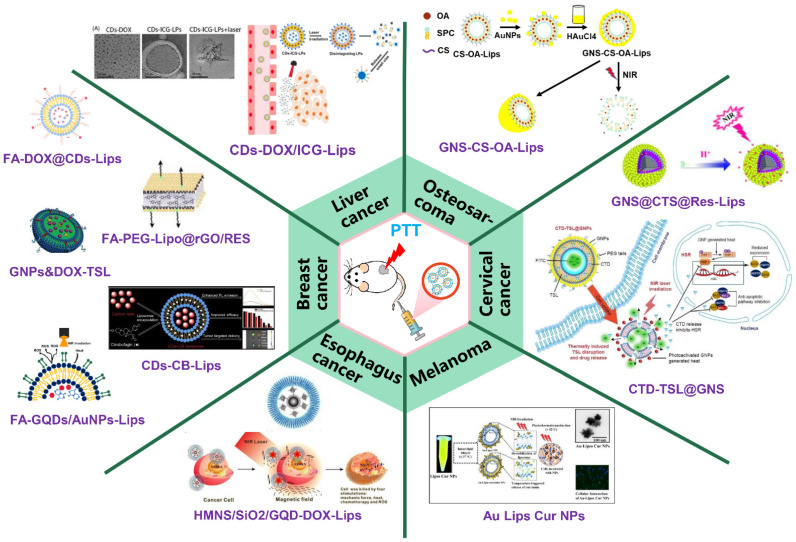
The recently fabricated NIRN-Lips for PTT of different types of cancer. Reprinted with permission from [9], John Wiley and Sons, 2016; [58], Elsevier, 2019; [67], Royal Society of Chemistry, 2019; [88], Dove Medical Press, 2018; [107] Royal Society of Chemistry, 2017; [118], Informa Healthcare, 2018; [119], Springer Nature, 2020; [120], Ivyspring International Publisher, 2016; [122], Springer Nature, 2020; [125], Royal Society of Chemistry, 2018; [141], Royal Society of Chemistry, 2017.

**Figure 7 pharmaceutics-13-02070-f007:**
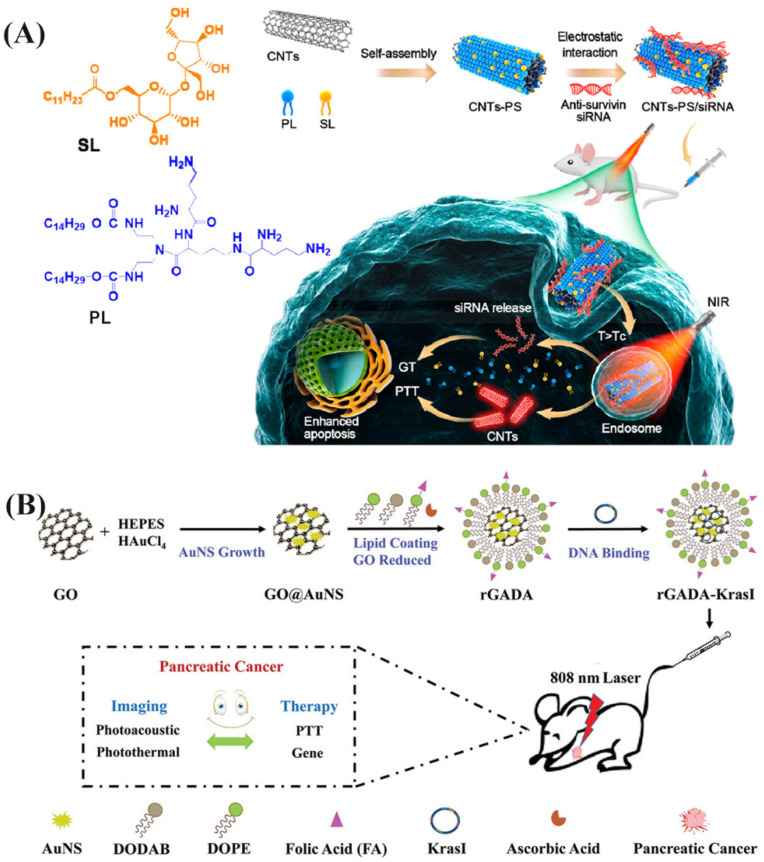
(**A**) Schematic diagram of the Temperature-Sensitive CNT-PS/siRNA Nanoparticle for Synergistic PTT and GT for Cancer Cells. Reprinted with permission from [36], American Chemical Society, 2021. (**B**) Schematic overview of the rGADA synthesis, and the rGADA/KrasI complexes used in gene/PTT synergistic therapy and photoacoustic/photothermal dual-modal imaging in pancreatic cancer. Reprinted with permission from [37], John Wiley and Sons, 2020.

**Table 1 pharmaceutics-13-02070-t001:** Different NIRN-Lips in terms of preparation method, encapsulation strategy, type of NIRNs and lipid composition.

Preparation Method	Encapsulation Strategy	Type of NIRNs	Lipid Composition	Drug Load	Average Diameter (nm)	Zeta Potential (mV)	Ref.
Thin-film hydration	Within the lipid bilayer	CNHs	DMPC, DSPC, Chol	/	80.0–100.0	/	[94]
	Within the lipid bilayer	CDs	DPHHP, DSHHP, Chol	CUR	128.0	29.9	[112]
	Within the lipid bilayer	AuNPs	SPC, Chol	PTX	281.1	45.3	[97]
	Within the lipid bilayer	QDs	EPC, Chol, DSPE-PEG2000	CGT	100.0	−17.1	[81]
	Within the lipid bilayer	QDs	PC, Chol, DSPE-PEG2000	APO	142.0	50.3	[96]
	Within the lipid bilayer	QDs	Chol, DSPE-PEG2000	siRNA	171.7	−2.7	[113]
	within the lipid bilayer	QDs	PC, Chol, PEG-6000	/	270.0	/	[114]
	Within the lipid bilayer	QDs	DPPC, DC-Chol, DSPE-PEG2000	/	89.7	20.1	[115]
	Within the lipid bilayer	QDs	L-α-lysolecithin, Chol	DOX	105.6	0.5	[116]
	Within the lipid bilayer	GO, CDs	DPPC, Brij 78, Chol	DOX	129.6	−7.3	[117]
	On the outer surface	AuNPs	SPC, Chol	DOX	100.0	−14.7	[68]
	On the outer surface	AuNRs	DOTAP, DOPE, Chol	NIR-797	89.0	46.4	[98]
	Encapsulated into the aqueous core	CDs	DSPE-mPEG2000, EPG, SPC, Chol	DOX	87.4	−12.9	[118]
	Encapsulated into the aqueous core	CDs	SPC, Chol, cephalin	/	80.0	−15.4	[55]
	Encapsulated into the aqueous core	CDs	DSPE-mPEG2000, DSPE-mPEG2000-FA, DOPE, DSPC, Chol	DOX	108.9	−31.4	[119]
	Encapsulated into the aqueous core	CDs	DSPE-mPEG2000, DPPC, Chol	CB	60.0–80.0	−2.6	[58]
	Encapsulated into the aqueous core	GO	SPC	DOX	391.3	/	[120]
	Encapsulated into the aqueous core	GNs	DPPC, Chol, DSPE-mPEG2000	DOX	141.0	−1.3	[121]
	Encapsulated into the aqueous core	AuNSs	P90G, Chol	calcein	170.0	−70	[99]
	Encapsulated into the aqueous core	AuNSs	DPPC, MSPC, DSPE-PEG-SH, Chol	PTX	293.9	2.5	[100]
	Encapsulated into the aqueous core	GQDs, AuNPs	DSPC, Chol	DOX	167.0	13.0	[122]
	Encapsulated into the aqueous core	AuNPs	EYPC, DSPE-PEG2000	VCR	113.4	−11.3	[123]
	Encapsulated into the aqueous core	AuNPs	SPC, Chol	TMZ	89.0	−69	[124]
	Encapsulated into the aqueous core	AuNPs	DSPE-PEG2000, DPPC	DOX	196.8	−29.5	[125]
	Encapsulated into the aqueous core	AuNPs	SPC, Chol, PEG2000	DOX	182.2	/	[101]
	Encapsulated into the aqueous core	AuNPs	DOPC, DOTAP, DSPE-PEG2000	VP	170.0	45.0	[126]
	Encapsulated into the aqueous core	AuNRs	SPC, HSPC, DSPE-PEG2000	DOC	163.1	−32.8	[127]
	Encapsulated into the aqueous core	QDs	DPPC, DSPG, DSPE-PEG2000	PTX	102.5	−19.8	[128]
	Encapsulated into the aqueous core	QDs	DSPC, DOTAP, Chol	/	114.0	24.8	[87]
	Encapsulated into the aqueous core	QDs	DOPC, DOPE	/	103.0	−13.2	[129]
	Encapsulated into the aqueous core/within the lipid layer	AuNPs	SPC, Chol, DSPE-PEG2000	PTX	149.2	−2.5	[130]
	Encapsulated into the aqueous core/within the lipid layer	AuNPs	DPPC	/	160.0	−6.2	[131]
	Encapsulated into the aqueous core/on the outer surface	AuNPs	DPPC, HSPC, EPC,	DOX	154.8	−38.0	[132]
	Encapsulated into the aqueous core/on the outer surface	AuNPs	DPPC, MPPC, DSPE-PEG2000	Calcein	118.0–146.0	−9.6	[133]
	Encapsulated into the aqueous core/on the outer surface	QDs	DSPC, DOTAP, DSPE-PEG2000	/	107.4	−0.8	[134]
	Encapsulated into the aqueous core/on the outer surface	QDs	DOPE, DSPC, Chol, DSPE-PEG2000	/	100.0	0	[135]
	On the outer surface	AuNPs	DPPC, HSPC, DSPE-PEG2000, Chol	CTD	96.4	28.7	[88]
	On the outer surface	GNS	SPC, Chol	RES	141.7	22.7	[107]
	On the outer surface	AuNPs	HSPC	CUR	100.0	22.0	[109]
	On the outer surface	AuNPs	DSPC, Chol	/	200.0	/	[136]
	On the outer surface	AuNPs	DSPC, Chol	QUE	120.0	11.8	[137]
	On the outer surface	AuNPs	SPC, CS	OA	172.0	22.7	[9]
	On the outer surface	AuNPs	SPC, Chol	BA	149.4	/	[138]
	On the outer surface	AuNPs	SPC	CUR	100.0–120.0	/	[67]
Solvent injection	encapsulated into the aqueous core	AuNPs	EPC, TPGS-COOH, Chol	DOC	217.1	−14.5	[89]
	Encapsulated into the aqueous core	AuNPs	EPC, DOPG	/	140.0–150.0	/	[139]
Ultrasonication	On the outer surface/encapsulated into the aqueous core	CDs	DSPC, Chol	/	230.0	20.0	[110]
	Within the lipid bilayer	GO	POPC	/	238.0	−15.2	[140]
	Within the lipid bilayer	GO	FA-PEG-DSPE, biotin-PEG-DSPE, DMPG	RES	148.0	−23.6	[141]
	Encapsulated into the aqueous core	AuNCs	DOPC, DSPE-PEG2000, Chol	TRP2	64.5	−10.0	[142]
Hydrothermal method	On the outer surface	CDs	triolein	/	103.0	/	[91]
Covalent attachment	On the outer surface	CNTs	SPC, DSPE-PEG2000, Chol	Oridonin	/	/	[111]
	On the outer surface	CNTs	biotin-PEG2000-PL, HSPC, PE	Calcein	10.0	−16.3	[92]
Ionic interaction assembly method	On the outer surface	GO	DPPC, Brij 78, DOTAP, Chol	DOX	153.9	−32.6	[57]
Plasmon resonance coating method	On the outer surface	AuCLs	DPPC, MPPC, DSPE-PEG2000	DOX	171.5	−1.0	[64]
Extrusion method	Encapsulated into the aqueous core	GNC	DOPC, N-dod-PE	/	175.0	−37.7	[90]
	Within the lipid bilayer	QDs	L-α-lysolecithin, Chol, PEG-Chol, DOPE	BP	104.2	−11.3	[143]

**Table 2 pharmaceutics-13-02070-t002:** Surface modified NIRN-Lips target to different tumor cells.

Type of NIRN-Lips	Drug Load	Surface Modification	Targeted Tumor Cells	Surface Engineering Techniques Used	Characterization	Ref.
FA-MWNTs-Lips	Oridonin	FA	HepG2 cells	FA-conjugated chitosan attached onto MWNTs-COOH using a non-covalent bond method; liposome containing oridonin covalently attached to MWNTs-COOH to form MWNTs-Lips.	FTIR, DLS, TEM, TGA	[111]
FA-CDs-Lips	/	FA	4T1 cells	Terminal amino functional group of CDs-Lips reacted with the carboxyl groups of FA.	DLS, TEM, FTIR	[110]
FA-GQDs/AuNPs-Lips	DOX	FA	4T1 cells	PEGylated FA (1 mg/mL) as targeting ligand was attached on the surface of AuNPs/QODs-Lips (5 mg/mL) through incubation process at room temperature.	FTIR, DLS, TEM, AFM, EDAX, X-ray, CT	[122]
FL/QDs-TK	/	FA	BEL-7402, Hep3B and SMMC-7721 cells	DSPE-PEG2000-folate were modified on liposomes by thin film hydration method.	DLS, TEM, FESEM, UV-vis, Bio-Rad imaging system	[115]
FA-DOX@CDs-Lips	DOX	FA	4T1 cells	DSPE-MPEG2000-FA was noncovalently inserted into the lipid bilayer.	DLS, FTIR, TEM, XPS, 1HNMR spectra	[119]
FA-PEG-Lip@rGO/RES	RES	FA, PEG	A549 and MCF-7 cells	0.1 μmol FA-PEG-DSPE was added to stabilize and modify liposome system.	DLS, TEM, AFM	[141]
CPP-CDs-Lips	CUR	CPP	MCF-7 cells	Carboxylic groups of CPP reacted with cholesterol to form conjugate.	DLS, TEM, FTIR	[112]
Man-CDs-Lips	/	D-mannose	HepG2 cells	D-mannose was non-covalently attached to the liposome surface.	TEM, AFM, XRD	[55]
SPACE-AuNSs-Lips	Calcein	SPACE peptides	NIH-3T3 cells	5 mg/mL POPE-NHS and 5 mg/mL SPACE peptide (pH = 8) were added into the mixture following a 2 h preincubation at room temperature.	DLS, TEM, DSC	[99]
DOC-AuGSH-TPGS-Tf	DOC	TPGS, Tf	glioma cells	TPGS-COOH on the liposome surface were activated and then incubated with 1 mL Tf solution (10 mg/mL) at room temperature for 30 min and kept overnight at 4 °C.	DLS, TEM, AFM, NMR	[89]
TPP-Lips-VP-10AuNPs	VP	TPP	mitochondria of HCT116 cells	DSPE-PEG2000-NH2 were inserted into the pre-formed liposomes, and then the PEGylated and TPP-coupled liposomes were prepared by the EDC-NHS coupling method.	DLS, TEM, spectrophotometer	[126]
Aptamo-QDs-Lips	siRNA	Anti-EGFR aptamer	MDA-MB-231 cells	DSPE-mPEG2000-aptamer were added to the prepared QDs-Lips and incubated for 4 h at 37 °C	DLS, TEM	[113]
Biotin-QDs-Lips	/	Biotin	A431 cells	Biotin-DSPE (0.012 μmol/mL) were added to prepare liposomes.	DLS, TEM, spectrofluorometer	[167]

Note: the full names of NIRN-Lips in Table 2 can be found in *Abbreviations*.

**Table 3 pharmaceutics-13-02070-t003:** Brief summary of the recently developed NIRN-Lips for cancer PTT.

Type of NIRN-Lips	NIR Laser	Temperature Reached	Drug Load	Antitumor Mechanism	Cancer Treated	Ref.
GNS-BA-Lips	808 nm	43 °C in 10 min	BA	(1) Local heat generated from NIR light cause PTT; (2) enhance intracellular BA accumulation	Cervical cancer	[138]
CDs-CB-Lips	500 nm	/	CB	(1) Increase cytotoxicity and cellular uptake of CB	Breast cancer	[58]
CTD-TSL@GNS	808 nm	44 °C in 20 min	CTD	(1) Block the heat shock response and inhibit the expression of HSP70 and BAG3, thus enhance therapeutic effect of CTD	Cervical cancer	[88]
Au Lips Cur NPs	780 nm	50 °C in 5 min	CUR	(1) Exert cytotoxic effect; inhibit cell proliferation and migration; (2) NIR light irradiation on Au Lips Cur NPs trigger the release of CUR	Melanoma	[67,109]
CDs-DOX/ICG-Lips	808 nm	56.8 °C in 5 min	DOX	(1) Induce cell apoptosis; (2) inhibit cell proliferation; (3) generate heat to kill cells	Liver cancer	[118]
FA-DOX@CDs-Lips	480 nm	/	DOX	(1) Induce cell apoptosis; (2) increase cytotoxicity and intracellular uptake of DOX	Breast cancer	[119]
HMNS/SiO2/GQD-DOX-Lips	808 nm	56.8 °C in 20 min	DOX	(1) Induce ROS generation and heat produced by NIR irradiation to kill cells	Esophagus cancer	[120]
FA-GQDs/AuNPs-Lips	750 nm	55 °C in10 min	DOX	(1) Generate ROS and heat to kill cells; (2) Increase cytotoxicity; (3) generate heat to kill cells	Breast cancer	[122]
GNPs and DOX-TSL; HGNPs and DOX-TSL	808 nm	45 °C in 5 min	DOX	(1) Increase cytotoxicity and cellular uptake of DOX; (2) transfer NIR light to heat	Breast cancer	[125]
DOX and HAuNS-TSL	808 nm	49.9 °C in 5 min	DOX	(1) Enhance cytotoxicity; increase intracellular DOX concentration	Liver cancer	[132]
DOX/AuCLs-TSL	808 nm	/	DOX	(1) AuCLs on the TSL absorb the NIR light to cause membrane destabilization; (2) increase cell cytotoxicity	Triple-negative breast cancer	[64]
AuNRs/DOCL-R	748 nm	60 °C in 10 min	DOC	(1) Enhance intracellular entrance; (2) increase DOC accumulation in tumor site; (3) induce ROS generation	Prostate cancer	[127]
QE-LipoAu	750 nm	48 °C in 7 min	QE	(1) After PTT, increase photothermal cytotoxicity, induce cell apoptosis; (2) depolymerize microtubules, suppress HSP70 expression, and cause DNA damage	Hepatocellular carcinoma	[137]
GNS-CS-OA-Lips	808 nm	/	OA	(1) After NIR light irradiation, a local temperature increase caused by AuNPs to kill cells; (2) hyperthermia promotes phase conversion from gel-to-liquid crystalline of cells membrane, and dramatically enhances intracellular uptake of OA, leading to the tumor cells apoptosis.	Osteosarcoma	[9]
FA-PEG-Lip@rGO/RES	780 nm	59.6 °C in 5 min	RES	(1) Enhance cellular uptake of RES; protect stability of resveratrol; (2) generate heat to kill cells	Breast cancer	[141]
GNS@CTS@RES-Lips	808 nm	66 °C in 10 min	RES	(1) Convert NIR light to heat to enhance the release and intracellular accumulation of RES	Cervical cancer	[107]
FA-CDs-Lips	808 nm	57–62 °C in 5 min	/	(1) Induce ROS generation; (2) increase cellular uptake	Breast cancer	[110]

Note: the full names of NIRN-Lips in Table 3 can be found in *Abbreviations*.

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
