# Peer review of "Design and Application of Near-Infrared Nanomaterial-Liposome Hybrid Nanocarriers for Cancer Photothermal Therapy"

_pharmaceutics, 2021, doi:10.3390/pharmaceutics13122070_

Round 1
Reviewer 1 Report
In this manuscript, the authors reviewed various methods to fabricate hybrid nanocarriers composed of near-infrared (NIR) responsive inorganic nanomaterials and liposomes that can be used for therapeutic purposes such as photothermal therapy. Authors listed characteristics of NIR responsive nanoparticles based on the core materials, size, shape, and surface charges and then introduced various strategies to load these nanoparticles into/onto liposomes. They also explicated the design principles in the incorporation process that can improve the performance of the hybrid nanocarriers. Then, authors covered mechanisms by which drugs are released by light after delivering them to the targeted sites in the form of the hybrid nanocarrier and elaborated applications including photothermal therapy. Lastly, the authors provide an outlook to further develop these hybrid nanocarriers by mentioning current challenges and limitations.
General Comments/Questions:
- Strength: this manuscript includes a comprehensive review of methods and excellent tables based on a large body of literature. Adding a figure or two that show representative data of exemplar papers would strengthen the manuscript.
- Some of the titles are not reflective of the contents. Sub-sections of each chapter might help to come up with proper titles.
- In Chapter 1, the authors provided a detailed explanation regarding the characteristics of NIR-responsive nanoparticles that are incorporated into the liposomes. However, how their properties improve the drug delivery by the hybrid platform with liposomes was not clear or too vague. This connection needs to be explicit in each of the subsections and throughout the manuscript.
- When introducing exemplar studies, authors should carefully choose relevant and important information to describe. Also, the description needs to be consistent throughout the manuscript.
- We strongly suggest an editorial service to improve grammar, wording, and the quality of sentences in the manuscript.
- The authors should go through the text and ensure that all acronyms are defined prior to use. The review as a whole has a lot of acronyms and jargon some of which are not spelled out. Spelling out the acronyms when first appearing is desirable. Plus, writing out the acronyms might improve readability if these acronyms are only used once or twice.
Specific Comments/Questions:
- In the abstract, mentioning "CT" in line 31 is out of the blue. First, CT needs to be spelled out. Second, the relation of CT to NIRN-Lip needs to be justified.
- Introduction
- In the first paragraph, the descriptions at cellular vs. organism levels are mixed, which confuses readers. These concepts need to be separately discussed.
- In the second paragraph, other types of external stimuli than light need to be introduced to justify the use of light as a tunable stimulator of the hybrid platform. This justification needs to include why near-infrared window is desirable.
- In chapter 1.1, G and GO need to be distinguished (and spelled out). Also, the relevance of GO to this review was unclear; so needs to be clarified.
- In chapter 1.3, NIR response of quantum dots and their relevance to photothermal therapy need to be specified. The current description is confusing because how semiconductor quantum dots respond to the NIR light is unclear. Further, authors mentioned that formulating the hybrid structure of quantum dots and liposomes provokes concerns at the end of the chapter. This is contradictory to many examples introduced in the manuscript including table 1. Authors need to justify/explicate why, despite many examples, this hybrid structure is concerning.
- In chapter 2.1, the seed-mediated method might be considered subordinate to the thin-film hydration approach. Authors might need to reconsider their classifications of the preparation method
- In chapter 2.2, discussions regarding the effect of size and shape are mixed up. These descriptions need to be separated.
- In line 435, PC are zwitterionic lipids, not cationic ones.
- In chapter 4.1, please reconsider the title. This chapter contains good examples, but does not clarify the mechanism of photothermal therapy. Also, the mechanism for photothermal therapy appears in part of chapter 4.1, for example, in line 696 (when mentioning ROS generation). This information needs to be consolidated in the same chapter.
- Consider adding references in lines 90, 171, 206.
Recommendation: reject and reconsider after clarification and addition of a summary figure with actual data
Author Response
Thanks for Reviewer 1's kind consideration. Based on your guidance, we have carefully revised the manuscript, which is shown in the attached word file.

Reviewer 2 Report
Here are my comments on the manuscript, I feel it is very comprehensive and should be accepted after dressing the following concernss.
- Pg1 line 31:please provide the full form of CT in the abstract also
- Pg2 line 7: please change localized surface resonance to surface plasmon resonance for better clarity and omit the following line.
- Some parts of the manuscript are too verbose, please try to change with crisp short sentences (page 2 line 101-112)
- Page 4 line 193: ‘single-walled carbon nanotubes (MWNTs)’ is probably incorrect
- The authors are suggested to kindly revise through the abbreviations (e.g. RES, SERS, CPP, etc) and their expanded forms, both the forms must be used in conjunction when mentioned for the first time in the manuscript for easy reference.
- The photothermal mechanism of NIRNs other than AuNPs is not clearly explained, please revise that part.
- The authors should incorporate recent original research articles (2021) in higher numbers to provide a more contemporary outlook to the already excellent review.
- Also check the abbreviation of single walled carbon nanotubes in the section of 1.1, line no. 143, as it is written as MWNTs, it seems to be SWNTs.
Author Response
Thanks for Reviewer 2's kind consideration. Based on your guidance, we have carefully revised the manuscript, which is shown in the attached word file.

Reviewer 3 Report
The review article “Design and application of near-infrared nanomaterial-liposome hybrid nanocarriers for cancer photothermal therapy” is well written and the authors have clearly mentioned the advantages/limitations and current developments on photo-thermal therapy of NIR-nanomaterials and liposomes based NIR-nanomaterials for cancer treatment. I recommend its publication after addressing the following comments.
Overall comments are as follows:
1. The article lacks citations at appropriate places.
2. Reference number(s) are missing at some places.
3. The authors can also include a short description of Iridium nanocrystals encapsulated liposomes for PTT therapy.
4. There are few grammatical errors in the article.
5. Few statements or the idea behind the statement is not very clear.
Therefore, authors are requested to thoroughly check the review article for possible errors before its re-submission.
Detailed comments are listed below:
Question or recommendation are mentioned in the bracket.
Line 37
……relies on chemotherapeutic drugs (reference)
Line 45
Among these nano-drug delivery systems (what does these refer to?)
Line 46
…….were widely studied as the potential drug delivery system in the early 1970s (reference)
Line50/51
Different drug molecules can be loaded into liposomes by hydrophilic-hydrophilic and hydrophobic-hydrophobic interaction (reference)
Line 61
….drug delivery and controlled drug release of liposomes (reference)
Line 84
…..efficacies of tumors (efficacies in tumors)
Line 85
…..localized surface resonance (surface plasmon resonance; SPR)
In Figure 1:
Gold nanoparticles (spherical gold nanoparticles)
Gold Nanostars and not Gold nanorod
Line 143
….single walled carbon nanotubes (MWCTs) ….(It should be SWCNTs and not MWCTs)
Line 148
……animal models have been reported (reference)
Line 151
…..2-5 nm in width and 40-50 nm in length (reference)
Line 155
….both NIR I and II areas (reference)
Line 162
…firstly reported in 2004 [38]…..reference citation is wrong as the year mentioned in reference is 2015, but it should be 2004.
Line 179
Because of RES can be adsorbed in the mesoporous CNTs (grammatically incorrect)
Line 186
….moderate high temperature (contradictory statement)…..if it is high, it can’t be moderate and vice versa.
Line 188
….to achieve synergistic treatment of tumors (reference)
Line 199
2.34 to 4.98 ± 0.59 nm [45]…(re-check reference number as this doesn’t belongs to Jiang et, al.)
Line 202
……to generate local heat (to generate localized hyperthermia)
Check reference number [48]… (it is not related to gold, rather it states magnetic nanoparticles)
Line 211
….lower LSPR extinction coefficient (reference)
Line 226
……distribution of drug molecules (reference)
Line 238
….other nanoparticles to attach (incomplete sentence)
Line 248
…fluorescence quenching and reduce its photostability (reference)
Line 256
of membrane structure (reference)
Line 258
and encapsulation efficiency of 96% (reference)
Line 265
high concentrations of calcium ions (reference number)
Line 298
were constructed to guide surgical resection by targeting glioma (reference)
Line 299
TEM and XPS confirmed that SPIONs and QDs were completely co-encapsulated (reference)
Line 301
liposomes with encapsulation rate of 88.9% (reference)
Line 308
…….confirmed by in vivo organs imaging (reference)
Line 314
stability of AuNPs and also showed high drug loading capacity (reference)
Line 319
with high structural stability and excellent optical performance (reference)
Line 327
advantages of rapid PTX release and easy intracellular uptake under hyperthermia (reference)
Line 333
the production of micelles may occur since the size of AuNPs over 6 nm (reference; re-write)
Line 337/338
After TEM images, AuNPs in the inner core of liposomes were clearly observed (statement not clear)
Line 341
loading CDs prepared by thin-film hydration (reference)
Line 350
CDs could also be enhanced with 4-folds (reference)
Line 354
CDs can be strongly affected by the interaction with polymer nanoparticles (reference)
Line 361/362
Figure 3: Captions are missing; label all parameters
Line 374
was prepared by loading RES into CTS modified liposomes, and coated with GNS (reference)
Line 380
…..gold iron?
Line 414
Reports have shown that only QDs with size less than 5 nm can be easily embedded in the hydrophobic part of the bilayer (reference)
Line 429
CNHs do not have the high aspect ratio problems associated with the toxicity that observed in longer CNTs (reference)
Line 484
increase retention and permeability at the tumor sites (reference number)
Line 491
gene-targeted cancer therapy (reference)
Line 507
average expression of normal cells (reference)
Line 509
Tf modified gold based theranostic liposomes has been reported for brain-targeted tumor therapy (reference)
Line 521
triple negative breast cancer (reference number)
Line 539
according to type of nanomaterials (reference number)
Line 559
chalcogenides and transition metal oxides, exhibiting higher light stability (rerefence)
Line 581/680
GNS@CTS@Res-lips …(initially it was written as GNS@CTS@RES-Lips)
……….drug release to minimize drug toxicity to normal cells (reference)
Line 621
……the cytotoxicity of PTT itself, improve the efficacy of secondary treatment (re-write)
Line 715
consisted of HAuNS and DOX (reference)
Line 772
NIRNs larger than 8 nm in size are difficult to be cleared through the liver and kidneys (reference)
Thanks
Regards
Author Response
Thanks for Reviewer 3's consideration. Based on your guidance, we have carefully revised the manuscript, which is shown in the attached word file.

Reviewer 4 Report
This manuscript is covering design, synthesis, drug release, applications, and future perspective of near-infrared nanomaterial-liposome hybrid nanocarriers (NIRN-Lips). Specifically, authors focus on application of NIRN-Lips for cancer photothermal therapy. Application of liposomes as a platform for photothermal therapy in combination with drug deliver and diagnostic imaging is increasing of interest in the last decade. This comprehensive review could be useful for the readers in the field of pharmaceutics, in particular liposomal drug delivery and controlled release. My review comments and suggestions for authors are as follows,
- The text in the Figure 1 is too small to read. The font size should be increased.
- Authors mainly focus on carbon-based nanomaterials, gold-based nanomaterials, and quantum dots as a photothermal material in this manuscript. On the other hand, photothermal dyes including indocyanine green are also major photothermal materials for liposome-based photothermal applications. It would be useful to discuss the advantage or disadvantage of photothermal nanomaterials based on carbon, gold, and quantum dots compared with molecular photothermal dyes.
- In NIR light-triggered Drug Release Mechanism section at page 14 and 15, authors discuss photothermal conversion mechanism of NIR light-activated nanomaterials (NIRNs) because most NIRN-Lips systems depend on photothermal effect that is achieved by the conversion of light into heat to induce liposome permeabilization. What is the actual temperature rise on the NIRNs? How can we measure the local temperature around liposomes in cells? Please discuss this point in the manuscript.
- In Table 3, near-infrared laser with 808 nm is mainly used for cancer PTT. What is the skin or tissue penetration depth of the 808 nm laser light? Is the tissue penetration of laser light considered a limitation to cancer treatment in clinical? Please discuss this point in the manuscript.
Author Response
Thanks for Reviewer 4's kind consideration. Based on your guidance, we have carefully revised the manuscript, which is shown in the attached word file.

Round 2
Reviewer 1 Report
manuscript can be accepted in present, revised form
Reviewer 2 Report
Accept in present form